# Prevention of post-traumatic stress disorder: Lessons learned from a terminated RCT of prolonged exposure

Maria Bragesjö[1]*, Filip K. Arnberg[2,3], Erik Andersson[1]

1 Division of Psychology, Department of Clinical Neuroscience, Karolinska Institutet, Stockholm, Sweden,
2 Department of Neuroscience, Psychiatry, National Centre for Disaster Psychiatry, Uppsala, Sweden,
3 Stress Research Institute, Stockholm University, Stockholm, Sweden

* maria.bragesjo@ki.se

**Data Availability Statement:** All relevant data are within the paper and its Supporting Information files.

## Abstract

The main purpose of the current trial was to test if a brief trauma-focused cognitive-behaviour therapy protocol (prolonged exposure; PE) provided within 72 h after a traumatic event could be effective in decreasing the incidence of post-traumatic stress disorder (PTSD), thus replicating and extending the findings from an earlier trial. After a pilot study (N = 10), which indicated feasible and deliverable study procedures and interventions, we launched an RCT with a target sample size of 352 participants randomised to either three sessions of PE or non-directive support. Due to an unforeseen major reorganisation at the hospital, the RCT was discontinued after 32 included participants. In this paper, we highlight obstacles and lessons learned from our feasibility work that are relevant for preventive psychological interventions for PTSD in emergency settings. One important finding was the high degree of attrition, and only 75% and 34%, respectively, came back for the 2-month and 6-month assessments. There were also difficulties in reaching eligible patients immediately after the event. Based on our experiences, we envisage that alternative models of implementation might overcome these obstacles, for example, with remote delivery of both assessments and interventions via the internet or smartphones combined with multiple recruitment procedures. Lessons learned from this terminated RCT are discussed in depth.

## Introduction

Psychologically traumatic events affect about 70% of the global population [1]. A clinically substantial proportion, an estimated 5.6% in Sweden, develop post-traumatic stress disorder (PTSD) [1], which includes symptoms of re-experiencing the event, avoidance, cognitive and mood changes, and hyperarousal [2]. PTSD is a detrimental condition and is associated with increased risk of suicide, drug and alcohol dependence, and sick leave [3] as well as with a higher prevalence of somatic problems including neurological, vascular, respiratory, gastrointestinal, and autoimmune diseases [4, 5].

**Funding:** The study was funded through The Swedish Research Council (grant 2016-02359), Swedish Society For Medicine (grant 658811) and Stockholm County Healthcare (grant 20170018). Neither of the funding organizations had any role in the conception of the study design or in the collection, analysis or interpretation of the data, in the writing of the report, or in the decision to submit the paper.

**Competing interests:** I have read the journal's policy and the authors of this manuscript have the following competing interests: F.K.A receives royalties from Natur och Kultur for the Swedish translation of the prolonged exposure treatment manual. E.A and M.B report no competing interests. This does not alter our adherence to PLOS ONE policies on sharing data and materials.

Despite the high societal and individual burdens associated with PTSD, only a fraction of cases are detected by the healthcare system [6], and evidence-based trauma-focused treatments for PTSD are seldom available in real-world settings [7, 8]. One way to decrease the prevalence of PTSD is to intervene before the disorder develops [9]. A pilot trial (N = 137) from the US by Rothbaum et al. [10] indicated that a modified version of prolonged exposure (PE; the first-line treatment for PTSD [11]) may be effective in preventing the onset of PTSD symptoms in emergency trauma care patients. The results from studies on early interventions are mixed. Studies on critical incident stress debriefing provided within the first days after the traumatic event indicates that the intervention is ineffective in preventing PTSD and may even worsen long-term symptoms of post-traumatic stress [12]. It has been suggested that these effects are due to trauma survivors being encouraged to talk about their experiences without having sufficient opportunity to emotionally process the traumatic event [12]. Other studies investigating 5–6 weeks of trauma-focused cognitive behaviour therapy starting within a couple of weeks after exposure to trauma as a treatment for acute stress disorder are more encouraging. Three trials have shown that trauma-focused cognitive behaviour therapy is more efficacious in reducing symptoms of post-traumatic stress compared to supportive counselling [13–15]. In the Rothbaum et al. [10] study, the PE intervention was initiated within 72 hours after trauma, followed by two weekly sessions. The authors motivated setting the time frame to 72 hours because it has been argued that failure of recovery after exposure to trauma (e.g. development of PTSD) may in part be explained as a failure of fear extinction [16], and animal research suggests that early extinction training has the potential to modify consolidation of the original fear memory [17]. In their trial, PE was superior to the assessment-only control group in reducing PTSD symptoms 12 weeks after the event [10], and the intervention seemed to reduce the risk for PTSD development [18]. The three-session protocol has more recently been evaluated against one session of intervention and an assessment-only comparator. No differences between the groups were found in the level of symptoms of post-traumatic stress, although the study was underpowered to detect between-group differences [19].

In 2016, our research group set out to replicate and extend the findings by Rothbaum et al. [10], originally with the aim to assess the effectiveness of the modified PE protocol in a larger sample and using an active control group of non-directive support [20, 21], blinded assessors, and longer follow up. After a pilot trial of 10 consecutive trauma patients, we subsequently launched a large-scale RCT. Given the estimated effect sizes in the previous trial by Rothbaum et al., we calculated that 352 participants (176 in both groups) were needed in order to detect a standardised effect size (Cohen´s $d$) of 0.3 (10% data attrition allowed, two tailed tests, $p = .05$, 80% power). However, a major reorganisation at the recruiting site posed such a large barrier to the recruitment that the RCT had to be terminated prematurely. In this paper, we describe the methods we used and the combined results from a total of 32 participants. We highlight different implementation issues of delivering PE for psychological trauma in patients within a hospital context and provide suggestions on how to resolve these issues.

## Method

### Pilot and main RCT design

The pilot study of 10 participants and the subsequent large-scale trial used an identical randomised controlled design and procedures. Study participants were patients seeking medical care at an emergency department (ED) after experiencing a potentially traumatic event within the past 72 hours. Participants were randomly allocated to either modified PE or a control condition. The sample reported throughout this paper is the pooled sample of 10 participants from the pilot study and the first 22 participants from the RCT.

## Participants

Eligible participants were Swedish-speaking patients over 16 years of age attending the ED at Karolinska University Hospital in Solna, Sweden, within 72 hours after experiencing a psychologically traumatic event according to the DSM-5 criterion A for PTSD (i.e., exposed to death, threatened death, actual or threatened serious injury, or actual or threatened sexual violence). Exclusion criteria were a) ongoing intoxication (e.g. due to alcohol or other drugs), b) low cognitive capacity, and c) other serious psychiatric comorbidity (ongoing psychotic symptoms or ongoing manic episode, suicidal ideation). Common traumatic events that lead to attendance at the ED were motor vehicle accidents and assaults.

## Procedure

The recruitment included several steps. A pre-selection screening was done each weekday morning at an out-patient clinic nearby the ED by a clinical psychologist who scanned the digital medical records of newly arrived patients in order to assess criterion A and potential exclusion criteria. Potentially eligible patients were then assessed on the inclusion and exclusion criteria using a structured assessment interview developed for the purpose. As an aid in conducting the assessment, the self-rating questionnaires Immediate Stress Response Checklist [ISRC; 22] and Montgomery Åsberg Depression Rating Scale–Self-rated version [MADRS-S; 23] were included. The assessment was conducted by one of four clinical psychologists involved in the study team either in the emergency room or at bedside in the hospital ward. In the measures section, a more detailed description of the trauma assessment procedures can be found. Patients who had been discharged before screening were interviewed via telephone and, if eligible, invited to the hospital to receive the intervention in the office of the clinical psychologist. Eligible patients received both written and verbal information about the study. After the eligible patients had signed the informed consent form, the psychologist conducted the baseline assessments and subsequently opened the sealed envelope containing the randomisation assignment. The intervention was delivered starting immediately after randomisation. Participants in both groups were assessed at weeks 1–3 (during the intervention) and at 2 months and 6 months after the intervention. The primary endpoint was 6 months. The participants were not reimbursed for their participation in the study. The study started recruitment on 18 April 2017 and was terminated on 11 November 2017. All included individuals continued the intervention and assessment procedures, even after the recruitment was terminated, as stated in the informed consent.

## Measures

The primary outcome was PTSD symptom severity assessed using the Clinician-Administered PTSD Scale for DSM-5 (CAPS-5) [24, 25]. Before the interview, the participants filled out the self-report measure Life Events Checklist for DSM-5 (LEC-5), which screens for potentially traumatic events in a respondent's lifetime. The incident leading to the ED visit was defined as the index traumatic event during the CAPS-5 interview. One participant who reported distress due to exposure to another previous traumatic event was scheduled for a separate visit to assess symptoms due to this particular event. The assessors (n = 4) were clinical psychologists who were experienced in conducting structured diagnostic interviews. Each assessor had received extensive training, up to 2 ½ days, in the administration and scoring of the CAPS-5. Inter-rater agreement was excellent (Intra-Class Correlation between 0.91 and 0.98 on the different assessment test sessions). Supervision was provided for difficult cases. Secondary outcomes were the PTSD Symptom Checklist for DSM-5 (PCL-5) [26], an intrusion diary [27, 28],

MADRS-S [23], the World Health Organization Disability Assessment Schedule-12 (WHO-DAS) [29], EQ-5D (Euroqol-5D) [30], and the Insomnia Severity Index (ISI) [31].

Symptoms of post-traumatic stress were assessed by the PCL-5 [26], a self-report measure assessing the 20 DSM-5 symptoms of PTSD. Participants were instructed to use the incident leading to the ED visit as the index event when completing the PCL-5. The participants also completed an intrusion diary, which assessed the daily number of intrusive memories of the traumatic event from day one of the intervention to the last session. This intrusion diary has been used in previous early intervention studies [27, 28] and was translated into Swedish by the first author.

Depressive symptoms were assessed using the MADRS-S [23], general functioning was assessed using the WHODAS-12 [29], quality of life was assessed using the EQ-5D [30], and sleep quality was assessed using the ISI [31]. Participants who were unable to attend the follow-up assessments at the clinic were interviewed via telephone.

The frequency of adverse events was assessed at session two and three of the intervention and at 2 and 6 months' follow up using a structured interview. To control for possible bias due to expectation and non-specific factors between the PE and control group, we used the Treatment Credibility Scale [32] at session 1.

## Interventions

Participants in both conditions received a three-session intervention in which the first session was provided immediately after the screening, which took place less than 72 hours after the traumatic event. The therapists were five clinical psychologists with extensive experience and training in PE. All therapists received an additional 1-day training in the adapted PE protocol used in this study as well as how to deliver the control intervention. Supervision was provided when needed.

**Prolonged exposure (PE).** The brief PE intervention protocol from the Rothbaum et al. [10] study was provided to us by the study authors. The protocol was translated into Swedish by the first author, who has been trained and certified in PE by the original treatment developer (Professor Edna Foa) and who, prior to the study, worked clinically with PE for 17 years. PE is based on the emotional processing theory developed by Foa and Kozak [33]. This theoretical framework posits that PTSD symptoms reflect pathological fear structures that do not accurately represent reality and are seen as signs that the traumatic memory has not been sufficiently emotionally processed. This theory stipulates that in order to achieve emotional processing of the trauma memories, the patient needs to repeatedly activate the fear structure and access corrective information about the trauma memory. A failure in emotional processing is conceptualised to occur due to avoidance of the memory itself and its associated feelings, thoughts, and situations related to the trauma, which limits the activation of the fear structure and access to corrective information. These avoidance behaviours are thought to maintain the individual's erroneous negative perceptions about themselves and the world. PE aims to inhibit avoidance patterns by gradually approaching the trauma-related stimuli, which is hypothesised to activate the fear structure and subsequently provide corrective information about the feared consequences of trauma. The adapted PE intervention used in this study has the same basic aim as traditional PE for PTSD with the difference that this early intervention has the potential to more swiftly modify the consolidation of the fear memory from the traumatic event [34].

In the first session, the participant is first given a brief rationale for the modified PE protocol and the function of avoidance behaviours for trauma memories. The rationale is followed by imaginal exposure in which the participant is instructed to re-activate the trauma memory

(i.e., to visualise the event in their mind's eye) for 20–30 minutes by recounting the traumatic event, talking in the present tense, and repeating the memory if necessary to reach the time limit. Subjective units of distress (SUD) are assessed regularly during each imaginal exposure. After the imaginal exposure, reflective questions are used to address erroneous trauma-related cognitions. The imaginal exposure is recorded on a voice recorder, and the participant is instructed to listen to the session on a daily basis in order to achieve emotional processing of the trauma memory.

In addition to imaginal exposure, the participant is also given psychoeducation about common reactions after trauma and the importance of breaking avoidance patterns of situations that trigger trauma memories. Exposure in vivo is carried out through a homework assignment. The participant also is given basic skills in a breathing retraining technique. Sessions two and three follow the same format with continued imaginal and in-vivo exposure. The intervention ended at session three with a summary of what the participant had learned from the sessions. All sessions were 60 minutes long and were conducted on a weekly basis.

**Control group.**   The main purpose of the control group was to provide a credible intervention that is in line with international recommendations on early interventions after trauma [20, 21], including psychoeducation about common reactions after trauma and general non-directive support aimed at promoting safety, calmness, connectedness, self-efficacy, and hope. The sessions followed the same structure and were matched to the intervention in frequency and duration. In the sessions, the therapist usually discussed, in a supportive way, how the week had progressed for the participant, but participants were free to address any issues at each session. If the participants talked about experiencing symptoms of post-traumatic stress, the therapist would normalise the reactions, provide emotional and practical support, calm the participant if necessary, and try to instil hope. The participants were free to discuss their traumatic event with their therapist; however, the therapist did not provide any rationale and/or suggestions for exposure exercises.

## Randomisation and masking

Block randomisation was conducted by an independent party (Karolinska Trial Alliance) in a 1:1 ratio. The participants were randomised after the baseline assessment using sealed envelopes. The sealed envelopes were otherwise kept safe and accessible only to persons authorised to unblind, and the randomisation key was kept at the clinic. Participants were blinded to the conditions in order to control for expectation confounders. Therefore, participants were simply told that they would receive one of two interventions aiming to prevent the onset of PTSD.

## Ethics approval and consent to participate

The study was registered at Clinicaltrials.gov (ID: NCT03116165) and approved by the Regional Ethical Review Board in Stockholm, Sweden (ID: 2015/1820-31). The study is reported in accordance with the CONSORT statement for nonpharmacological treatments.

Eligible patients received both written and verbal information about the study and were not included in the study until after signing the written informed consent.

## Results

### Study progression

Unforeseen major organisational changes at the hospital negatively affected the recruitment. At the start of the RCT, 30% of the trauma patients at the ED had been recruited into the current study. This proportion decreased rapidly to only 2.8% after the organisational changes in

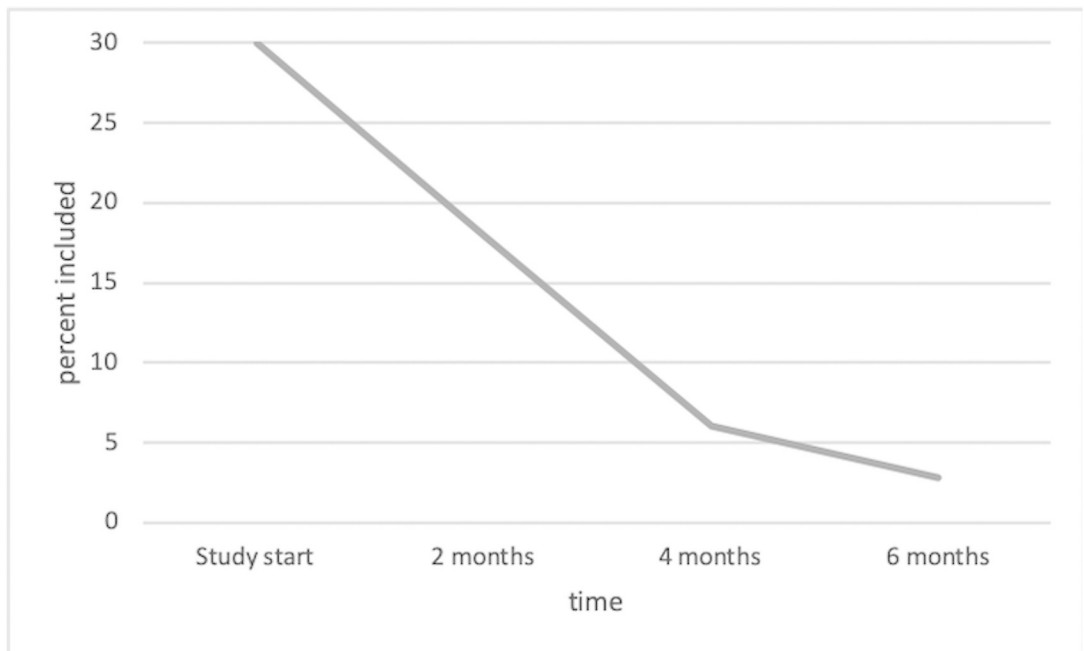

**Fig 1. Recruitment rate of patients attending the ED during the study period.**

the subsequent months (illustrated in Fig 1). All trauma care was moved to an entirely new hospital, Karolinska University Hospital, which was designed for highly qualified specialised trauma care. The hospital was now mandated to take on only very severely injured patients, leading to a markedly lower number of patients overall and a lower proportion of eligible patients for the study. The recruitment continued for 6 months because there was uncertainty as to whether the decline in eligible patients would become permanent, but the low inclusion rate due to the organisational changes ultimately forced the discontinuation of the study. Treatment adherence and credibility were rated as high, whereas many participants dropped out at the follow-up assessments.

The flow of participants is shown in Fig 2. There were 292 trauma patients who were screened using the digital screening record, and 32 participants were included in the study. The most common reasons for exclusion were not fulfilling criterion A (n = 71), not being able to contact the patient (n = 41), and that more than 72 hours had already passed since the traumatic event (n = 38). Twenty-five participants had experienced an accident and seven were victims of assault. On average, a mean of 35 hours had passed from the traumatic event to the intervention. All but one of the participants had slept before the intervention.

There were gender differences between the included participants (12 men and 20 women) and patients who were eligible but declined participation (31 men and 8 women). Younger people were also more likely to decline participation in the study, and the included patients had a mean age of 42 years while those patients who declined participation were on average 32 years old. Baseline characteristics for the included participants are presented in Table 1.

## Preliminary outcome data

SUD ratings for all three intervention sessions are shown in Fig 3. The CAPS-5, PCL-5, and MADRS-S scores are shown in Table 2. The participants reported no adverse events that could be attributed to either of the interventions.

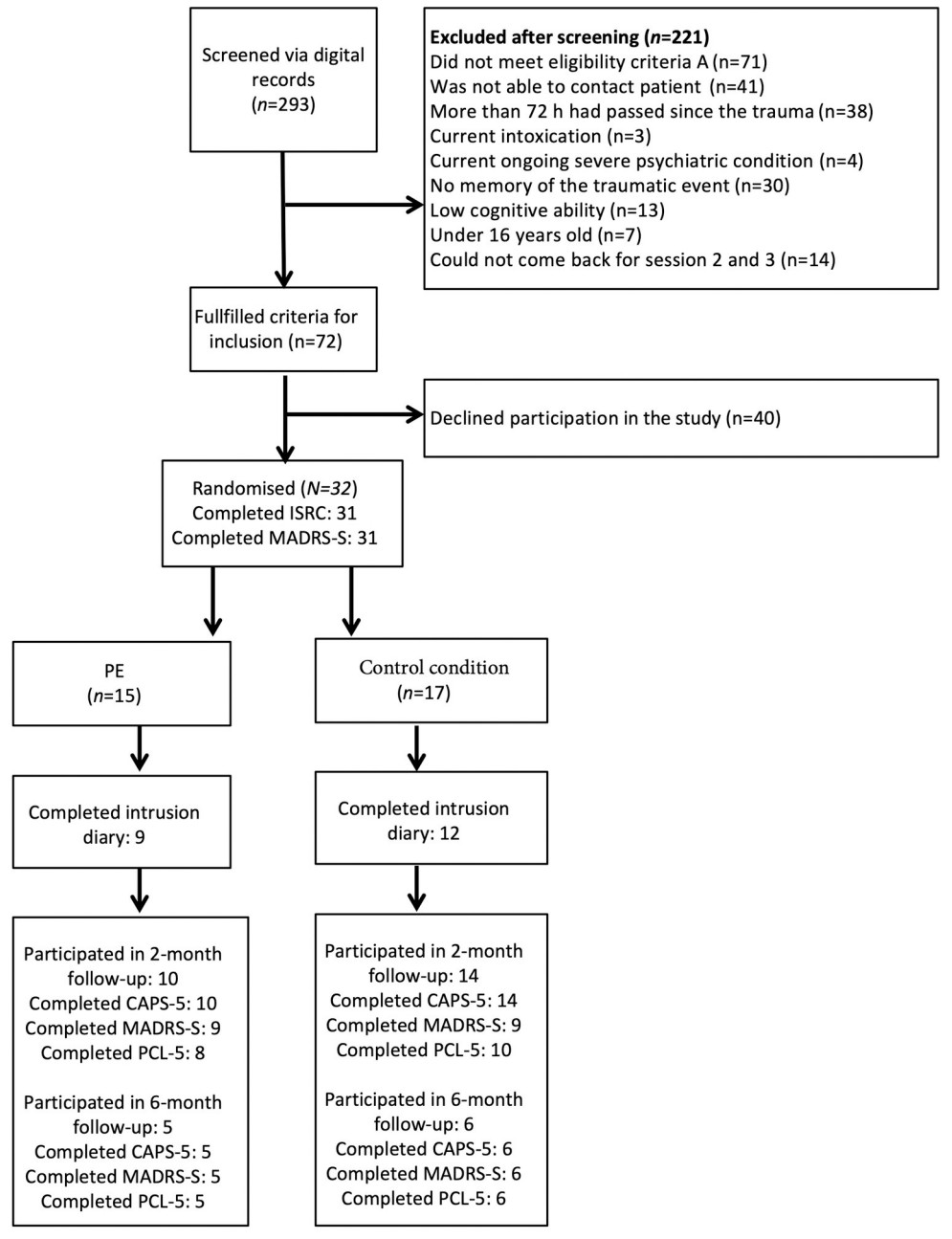

**Fig 2. Participant flow chart.**

The scores on the Treatment Credibility Scale were overall high in both conditions ($M_{PE}$ = 41.2, $M_{control}$ = 37.5). Four participants, two from each condition, dropped out of the intervention. One participant perceived that being randomised to the control condition was negative and declined further participation. The other participant in the control condition did not give a reason for leaving the study. One participant in the PE group left the study and stated that it was due to them perceiving the intervention as too emotionally demanding. The other participant in the PE group stated not needing the intervention as the reason for dropping out.

**Table 1. Baseline characteristics for included participants by intervention condition.**

| | Prolonged exposure (n = 15) | Control condition (n = 17) |
|---|---|---|
| **Gender** | | |
| Women, n (%) | 9 (60%) | 11 (64%) |
| **Age (years)** | | |
| Age, mean (SD) | 41 (13) | 44 (10) |
| **Occupational status** | | |
| Working | 14 (93%) | 14 (82%) |
| Student | 0 | 1 (6%) |
| On sick leave | 0 | 1 (6%) |
| Parental leave | 0 | 1 (6%) |
| Unemployment | 0 | 1 (6%) |
| **Traumatic event** | | |
| Accident, n (%) | 11 (73%) | 14 (82%) |
| Assault, n (%) | 4 (27%) | 3 (18%) |
| Time since traumatic event, mean hours (range) | 37 (4.5–57) | 33.5 (12.5–71) |
| Admitted as in-patient, n (%) | 3 (20%) | 3 (18%) |
| **History of trauma and mental illness** | | |
| Prior exposure to trauma as an adult, n (%) | 6 (40%) | 6 (35%) |
| Prior exposure to trauma as a child, n (%) | 6 (40%) | 4 (23.5%) |
| Previous or current mental illness, n (%) | 8 (53%) | 8 (47%) |
| **Baseline measures** | | |
| ISRC score, mean (SD) | 28.5 (12.9) | 26.9 (12.5) |
| MADRS-S score, mean (SD) | 15.6 (9.6) | 16.5 (11.6) |

Abbreviations: ISRC, Immediate Stress Response Checklist; MADRS-S, Montgomery Åsberg Depression Rating Scale, Self-rated

A total of 24 of the 32 enrolled participants (75%) completed the CAPS-5 assessment at the 2-month follow-up, and 11 (34%) completed it at the 6-month follow-up. Seven of the 24 assessments (29%) at the 2-month follow-up were conducted by telephone. We saw a small gender difference in who attended the 6-month follow-up assessment–with 30% (6/20) of the recruited women returning and 41% (5/12) of the recruited men returning.

Nine (60%) participants in the PE group completed the intrusion diary between sessions 1 and 2, and 7 (47%) completed it between sessions 2 and 3. The corresponding figures for the

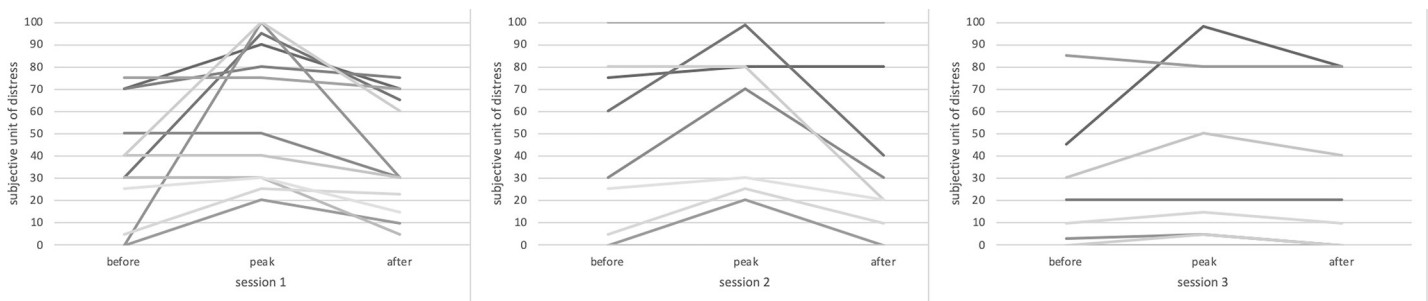

**Fig 3. Individual subjective units of distress ratings across the intervention sessions.** A decrease in the participants' mean subjective level of distress during the intervention is seen (session 1 –pre-SUD 35, peak-SUD 61, post-SUD 40; session 2 –pre-SUD 42, peak-SUD 56, post-SUD 33; session 3 –pre-SUD 24, peak-SUD 33, post-SUD 28).

**Table 2. Treatment outcomes for included participants by intervention condition.**

| Outcomes | Prolonged exposure | | Control condition | |
| --- | --- | --- | --- | --- |
| | N | *mean* (SD) | N | *mean* (SD) |
| **Primary outcome CAPS-5** | | | | |
| 2-months | n = 10 | 17.9 (15.7) | n = 14 | 13.1 (15.3) |
| 6-months | n = 5 | 4.0 (4.4) | n = 6 | 9.5 (11.8) |
| **Secondary outcomes** | | | | |
| PCL-5 | | | | |
| post-intervention | n = 9 | 20.4 (13.9) | n = 13 | 24.8 (20.2) |
| 2-months | n = 8 | 17.8 (15.2) | n = 10 | 13.4 (16.8) |
| 6-months | n = 5 | 7.3 (5.6) | n = 6 | 8.6 (5.3) |
| MADRS-S | | | | |
| 2-months | n = 9 | 11.9 (10.3) | n = 9 | 9.7 (13.7) |
| 6-months | n = 5 | 7.6 (15.3) | n = 6 | 13.7 (16.6) |
| **Baseline measures** | | | | |
| ISRC sum score | n = 14 | 28.5 (12.9) | n = 17 | 26.9 (12.5) |
| MADRS-S sum score | n = 15 | 15.6 (9.6) | n = 16 | 16.5 (11.6) |

Abbreviations: CAPS-5, Clinician-Administered PTSD scale for DSM-5; PCL-5, Posttraumatic stress disorder checklist for DSM-5; MADRS-S, Montgomery Åsberg Depression Rating Scale, Self-rated; ISRC, Immediate Stress Response Checklist

control condition were 12 (70%) between sessions 1 and 2 and 9 (53%) between sessions 2 and 3. SUD ratings collected during the imaginal part of the PE sessions were available from 13 (86%) participants.

The mean numbers of intrusive memories are shown in Fig 4. The mean number of daily intrusions between sessions 1 and 2 was 4.68 in the PE group and 4.68 in the control condition. Between sessions 2 and 3, the mean number of daily intrusions was 0.19 in the PE group and 2.54 in the control condition.

## Discussion

The current study was designed as a preventive intervention for PTSD to replicate and extend the Rothbaum et al. [10] trial. After a successful pilot trial, we aimed to recruit 352 patients at the ED in an RCT to test the effects of modified PE compared with an active control condition.

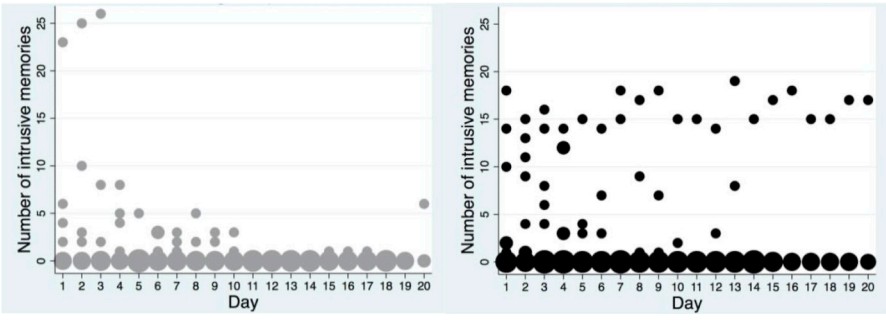

**Fig 4. Frequency scatter graphs of the number of intrusive memories per day recorded in the intrusion diary during the intervention period for patients who returned the diary in each condition (the left graph depicts the PE group and the right the control condition).** The circle size illustrates the number of participants who reported the indicated number of intrusive memories for each day.

The RCT started out with a fair recruitment rate of 30% of attending patients but had to be discontinued after the hospital underwent unforeseen major organisational changes and started to treat only severely injured patients, which led to a drastic decline in the number of eligible patients that could be recruited into this study.

In this study, we took advantage of already established procedures and infrastructure at the hospital. For example, we used regular hospital psychologists and the hospital´s medical record system at the ED department to find and approach eligible trauma patients. We consider this regular clinic approach to be a major strength in the study because it would have enhanced implementation after study completion. The reliance on personnel and infrastructure at one specific hospital unfortunately became a major obstacle in this study because recruitment rates fell significantly during the re-organisation. In hindsight, we might have circumvented the problem with recruiting patients if we had developed study procedures that were able to function independently of regular hospital routines and organisational changes within the hospital. The current study also had rather high levels of attrition.

Another option for future research to overcome the problems mentioned above would be to recruit, assess, and provide trauma patients with early interventions remotely, for example, by using smartphone or internet-delivered approaches. Studies on internet-based CBT have shown promising results for the treatment of PTSD [35, 36]. This approach could ensure swift and easy recruitment, assessment, and provision of interventions as well as facilitate stability in study procedures independent of the current hospital organisation. Internet-delivered psychological treatments can also increase accessibility when implemented in regular care [37]. An interesting future innovation would therefore be to investigate if it is possible to reach trauma patients remotely in the early aftermath of traumatic events. As previously mentioned, an important aspect in this current trial was the high degree of attrition. Many individuals recruited in this trial were not PTSD cases and may therefore not have been motivated to undergo any more follow-up assessments because it might have been inconvenient (e.g., traveling to the hospital, navigating to a new clinician, or missing work). Remotely delivered and easily accessible technological tools could potentially circumvent some of the attrition found in this study. However, one large-scale trial from 2013 found relatively low compliance rates for an automated internet-based psychological preventive intervention for PTSD [38]. Future trials could add therapist support to internet-delivered interventions because this has been shown to lead to a higher degree of compliance and improved efficacy [39].

A significant proportion of ED patients (24%) in the current study were excluded due to the time criterion. Many patients had already returned home after their visit to the ED when we detected them in the medical record system, and they could thus only be scheduled for treatment the day after the traumatic event. Memory consolidation theory proposes that the optimal time to intervene before completed memory consolidation is within 6 hours after the traumatic event has taken place [34]. From that perspective, it is important to develop simple interventions that can be delivered to the patient at an early stage in the hospital, for example, as recently shown by Iyadurai et al. [27]. Furthermore, sleep has been hypothesised to play a role in memory consolidation [40], and thus the optimal timeframe for early interventions might be before the patient first sleeps after the event. In the current study, all but one patient had slept before the start of the intervention. Future research could investigate in more detail whether sleep has a moderating effect on early interventions for trauma.

Although memory consolidation theory proposes a narrow time window of 6 hours, this might not be clinically feasible for many patients experiencing trauma. Some patients are too physically ill to receive psychological interventions within this time frame, and other patients might be more motivated to undergo psychological interventions a few weeks post-trauma. The recruitment rates in three previous trials investigating early interventions after trauma

varied between 2% and 26% [10, 19, 27]. One opportunity to increase inclusion rates could therefore be to extend the time criterion of 72 hours to up to a week or more. A recent systematic review showed that, of 61 studies investigating the efficacy of early interventions for trauma, the majority of these studies investigated interventions that were provided within weeks or even months after exposure to trauma. The conclusions drawn from this review point to evidence for the provision of early interventions provided only to individuals with impairing symptoms rather than to everyone who recently experienced trauma [41]. One suggestion would thus be to provide early PE to individuals who experience significant intrusive memories or other symptoms that can be readily addressed with existing interventions. This indicated approach seems to have potential to be more effective because individuals with more symptoms also tend to be more motivated to complete both treatment and assessment procedures.

To summarise, this prematurely terminated trial generated important scientific and clinical experiences. Based on these experiences, we suggest that future research into the prevention of PTSD might benefit from considering implementation models with remotely delivered, easily accessible intervention and assessment procedures that are independent of regular health organisation routines, as well as simple and easily delivered interventions provided by non-specialists at the ED to patients with a certain level of symptoms of post-traumatic stress. In addition, future studies of early interventions may need to evaluate the potential benefits of a very short time-frame for inclusion against the difficulties in recruitment and attrition that arise from such a narrow time-frame.

## Limitations

A major limitation with the study is the small sample size and limited follow-up data. The data from the pilot study and the RCT were combined, which might have introduced excess variability in the data. However, in this trial the pilot phase did not lead to any significant changes in the study procedure and thus the variability introduced would be negligible. Nonetheless, due to the small sample size the preliminary outcome data were not subjected to statistical analysis and should be regarded as tentative. Furthermore, the termination of this trial was due to organisational changes specific to the study context, and the significance for other trials should be interpreted with caution. Similarly, the participants were not assessed in depth for current PTSD due to other events or psychiatric comorbidity. Nonetheless, the difficulties with low recruitment rates and high attrition rates at follow-up assessments have been noted by other trials of early intervention as well [10, 19, 27]. We suggest therefore that this terminated trial contributes to a growing body of literature that indicates important lessons on the feasibility aspects of conducting early intervention trials of psychological interventions for trauma victims.

## Supporting information

**S1 Checklist. Consort guideline checklist for the study.**
(DOC)

**S1 File. Baseline data.**
(DOCX)

**S1 Data. Follow up data at 2 months.**
(XLSX)

**S2 Data. Follow up data at 6 months.**
(XLSX)

## Acknowledgments

The authors express their sincere gratitude to Emily Holmes who provided a significant intellectual contribution to the design of this study.

## Author Contributions

**Conceptualization:** Maria Bragesjö, Filip K. Arnberg, Erik Andersson.

**Data curation:** Maria Bragesjö, Filip K. Arnberg, Erik Andersson.

**Formal analysis:** Maria Bragesjö, Filip K. Arnberg, Erik Andersson.

**Funding acquisition:** Maria Bragesjö, Filip K. Arnberg, Erik Andersson.

**Investigation:** Maria Bragesjö, Filip K. Arnberg, Erik Andersson.

**Methodology:** Maria Bragesjö, Filip K. Arnberg, Erik Andersson.

**Project administration:** Maria Bragesjö, Erik Andersson.

**Supervision:** Filip K. Arnberg, Erik Andersson.

**Validation:** Filip K. Arnberg.

**Writing – original draft:** Maria Bragesjö, Erik Andersson.

**Writing – review & editing:** Maria Bragesjö, Filip K. Arnberg, Erik Andersson.

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
