## [Decision Letter · Decision Letter 0]

17 Jun 2020

PONE-D-20-01989

Prolonged Exposure as early intervention in an emergency department context: lessons learned from a terminated RCT.

PLOS ONE

Dear Dr. Bragesjö,

Thank you for submitting your manuscript to PLOS ONE. After careful consideration, we feel that it has merit but does not fully meet PLOS ONE’s publication criteria as it currently stands. Therefore, we invite you to submit a revised version of the manuscript that addresses the points raised during the review process.

We look forward to receiving your revised manuscript.

Kind regards,

Natasha McDonald

Associate Editor

PLOS ONE

Journal Requirements:

2. Please ensure you have thoroughly discussed any potential limitations of this study within the Discussion section as only one limitations is mentioned.

3. Your ethics statement must appear in the Methods section of your manuscript. If your ethics statement is written in any section besides the Methods, please move it to the Methods section and delete it from any other section. Please also ensure that your ethics statement is included in your manuscript, as the ethics section of your online submission will not be published alongside your manuscript.

4. Please include captions for your Supporting Information files (CONSORT checklist Lessons learned .doc file changed from "other" to "supporting information" item type) at the end of your manuscript, and update any in-text citations to match accordingly. Please see our Supporting Information guidelines for more information: http://journals.plos.org/plosone/s/supporting-information

'E.A and M.B report no competing interests.

I have read the journal's policy and the authors of this manuscript have the following competing interests: F.K.A receives book royalties from Natur och Kultur for the Swedish translation of the prolonged exposure treatment manual.'

Reviewers' comments:

Reviewer's Responses to Questions

**Comments to the Author**

1. Is the manuscript technically sound, and do the data support the conclusions?

Reviewer #1: No

Reviewer #2: Yes

2. Has the statistical analysis been performed appropriately and rigorously? 

Reviewer #1: N/A

Reviewer #2: Yes

3. Have the authors made all data underlying the findings in their manuscript fully available?

Reviewer #1: No

Reviewer #2: No

4. Is the manuscript presented in an intelligible fashion and written in standard English?

Reviewer #1: Yes

Reviewer #2: Yes

5. Review Comments to the Author

Reviewer #1: Important note: This review pertains only to ‘statistical aspects’ of the study and so ‘clinical aspects’ [like medical importance, relevance of the study, ‘clinical significance and implication(s)’ of the whole study, etc.] are to be evaluated [should be assessed] separately/independently.

Since this paper highlight obstacles and lessons learned from a terminated RCT / feasibility work relevant for preventive psychological interventions for PTSD in emergency settings, the title should indicate it (inclusion of word PTSD is must), in my opinion. Present title gives the impression that the paper is about methodology ‘regarding how to learn lessons from any terminated RCT’ in general. Look at the ‘objectives’ given in line 25-28.

This RTC is terminated even before recruiting 10% of targeted (minimum required) sample size of 352 participants [32/352=9.09% or 22/352=6.25%], so that the lessons learned from this terminated RCT data are not much meaningful. As said in lines 35-37 “One important finding was the high degree of attrition: only 78% and 34% respectively came back for the two months and six-months assessments” does that mean that although recruitment stopped, data collection continued?

Is it correct to combine data from the pilot study and whatever available from terminated RCT? as said in lines 81-83 [The sample reported throughout this paper is the pooled sample of 10 participants from the pilot study and the first 22 from the RCT]. Please check typing error in sentence in lines 117-8 that “Inter-rater agreement was excellent (Inter Class Correlation between 0.91-0.98).” which should be “Inter-rater agreement was excellent (Intra-Class Correlation Coefficient between 0.91-0.98).” because we assess Inter-rater agreement by Intra-Class Correlation Coefficient (and not by Inter Class Correlation). Further, note that there is only one value of Intra-Class Correlation Coefficient and a range as given. If you intend to mean differently, please clarify.

I believe that ‘Procedures’ (given in lines 96-111) & subsequent account {like Measures, etc.} are part of ‘protocol’ [because the trial is terminated], in that case ‘The study started recruitment 18th of April 2017 and was terminated 11th November 2017’ (lines 110-11) means it took more than six months to recruit 22 subjects {or was that also planned?}. Check the ‘tense’ of the entire text.

In my considered opinion [despite the fact which is highly appreciable that tools used (or proposed to be used) are very appropriate] whatever authors want to communicate [definitely important to communicate], could be communicated in just a letter-to-editor or brief communication (and not a full-length paper/article).

Reviewer #2: This is an important study that describes an early intervention for PTSD. Although the study was not completed, the results are important in this field given the paucity of studies. My comments are as follows:

The results and methodology are not completely in sync in terms of previous psychological trauma and psychiatric disorder. How were these assessed? Were participants with current PTSD (ie relating to a previous event) excluded from the study? If not, why not?

Please provide more information regarding the follow up assessments - were those who attended follow up different from those who did not?

The authors mention sleep but this is not elaborated upon in the discussion given the literature regarding memory consolidation and sleep this is important.

The authors do not cover the general field of early intervention of PTSD. The results of the few studies that have been carried out in the first 72 hours, taken together, are not particularly encouraging. Most of the studies of early intervention have concluded that it is most effective with symptomatic survivors, not whole populations, and the few studies examining very early (ER based) interventions are not consistent. This is not reflected in the manuscript, neither in the introduction "One way to decrease the prevalence of PTSD is to intervene before the disorder develops" nor in the discussion, where the recommendation for the use of Internet based interventions or memory consolidation is made without examining the studies that do not support these results. The complexity of this issue should be discussed properly. Additionally the inclusion of participants more than 72 hours after the event raises questions about the use of memory consolidation that should be clarified.

The authors state that a number of reasons led to the premature end to the study, but only mention one.

6. PLOS authors have the option to publish the peer review history of their article (what does this mean?). If published, this will include your full peer review and any attached files.

Reviewer #1: No

Reviewer #2: No

---

## [Author Response · Author response to Decision Letter 0]

7 Aug 2020

PONE-D-20-01989

Journal Requirements:

1. We have updated the manuscript to meets PLOS ONE's style requirements.

2. We have thoroughly discussed and elaborated potential limitations of the study.

 3. Our ethics statement now appear in the Methods section of the manuscript. 

4. We have added captions for our Supporting Information files. Thank you for changing the file type for CONSORT checklist Lessons learned .doc file.

5. a. We have confirmed that our declarations of interest does not alter our adherence to all PLOS ONE policies on sharing data and materials, and included including the following statement: "This does not alter our adherence to PLOS ONE policies on sharing data and materials.” 

b. We have included the updated Competing Interests statement in our cover letter.

6. We have uploaded the minimal data set underlying the results as Supportive information files. Thank you for taking care of the update on our Data Availability statement. 

Dear Reviewer #1 and Reviewer #2,

First, we want to thank you for valuable comments. We have revised the manuscript accordingly and we think that it is greatly improved. Below follows a detailed description of the changes made point by point:

Reviewer #1

1. Since this paper highlight obstacles and lessons learned from a terminated RCT / feasibility work relevant for preventive psychological interventions for PTSD in emergency settings, the title should indicate it (inclusion of word PTSD is must), in my opinion. Present title gives the impression that the paper is about methodology ‘regarding how to learn lessons from any terminated RCT’ in general. Look at the ‘objectives’ given in line 25-28.

Response: We thank Reviewer #1 for this thoughtful comment and we have incorporated the word PTSD in the title of the manuscript (page 1, Prevention of Post-Traumatic Stress Disorder: Lessons Learned from a Terminated RCT of Prolonged Exposure).

2. This RTC is terminated even before recruiting 10% of targeted (minimum required) sample size of 352 participants [32/352=9.09% or 22/352=6.25%], so that the lessons learned from this terminated RCT data are not much meaningful. 

Response: We agree that the outcome data come with great uncertainty and we have elaborated on this further in the limitation section (page 18, first paragraph). As highlighted by Reviewer #1 in point 6 below, it took us nearly six months to recruit 22 participants. The study started out with a fair recruitment rate, 30% of the daily intake of trauma patients at the ED were included, but this figure declined swiftly to only 2.8% of trauma patients. This significant decline was due to major organizational changes at the recruiting site and was not expected. Although the number of included participants compared to the target sample was relatively low, we argue that the long recruitment period and sudden shift in recruitment rate provides the reader with important data on the role of organizational factors in early provided psychological interventions after trauma. Also, we do not judge 22-32 subjects as a meaningless sample size but sufficient to say something about the feasibility of using prolonged exposure as an early intervention in a hospital context. We agree with Reviewer #1 that statistical analysis of the sample may not be meaningful, and hence we have not conducted any such analyses. However, in line with Reviewer #2 below, we think the study still holds important lessons for both clinicians and researchers working with individuals early after exposure to trauma. 

3. As said in lines 35-37 “One important finding was the high degree of attrition: only 78% and 34% respectively came back for the two months and six-months assessments” does that mean that although recruitment stopped, data collection continued?

Response: Yes, this is correct, as recruitment stopped, we continued the follow up as stated in the verbal and signed informed consent with the included participants. We deemed this to be best ethical practice as participants expected to receive a clinician visit (with the possibility to be offered further treatment, if there would be any need, for mental health issues). We agree with Reviewer #1 that this needs further clarification and we have added information about this on page 5, last paragraph, continuing on page 6. 

4. Is it correct to combine data from the pilot study and whatever available from terminated RCT? as said in lines 81-83 [The sample reported throughout this paper is the pooled sample of 10 participants from the pilot study and the first 22 from the RCT]. 

Response: Because the trial was terminated early, we decided to include also the feasibility data from the pilot study. The participants in the pilot study underwent clinical procedures and routines that were identical to the participants in the RCT. Including participants also from this part of the larger study gave us greater precision to investigate feasibility aspects of using prolonged exposure as an early intervention in a hospital context. We think this improves the quality of the manuscript significantly but we are open to revise this if the Editor thinks differently. We have added a discussion of this under Limitations (page 18, first paragraph).

5. Please check typing error in sentence in lines 117-8 that “Inter-rater agreement was excellent (Inter Class Correlation between 0.91-0.98).” which should be “Inter-rater agreement was excellent (Intra-Class Correlation Coefficient between 0.91-0.98).” because we assess Inter-rater agreement by Intra-Class Correlation Coefficient (and not by Inter Class Correlation). Further, note that there is only one value of Intra-Class Correlation Coefficient and a range as given. If you intend to mean differently, please clarify. 

Response: We thank reviewer #1 for pointing us this typing error in the manuscript. The inter-rater agreement is assessed by the Intra-Class Correlation Coefficient. We have changed this in the manuscript (page 6, last sentence). We have also clarified that the range given is from multiple assessment sessions. 

6. I believe that ‘Procedures’ (given in lines 96-111) & subsequent account {like Measures, etc.} are part of ‘protocol’ [because the trial is terminated], in that case ‘The study started recruitment 18th of April 2017 and was terminated 11th November 2017’ (lines 110-11) means it took more than six months to recruit 22 subjects {or was that also planned?}. 

Response: We have structured the manuscript closely to standard reporting of RCT:s in order to facilitate the reading of the manuscript. We therefore include standard headings in the Methods section, although we do see that it might make sense to revise the structure. Nevertheless, we would suggest to keep the current structure. As to the question from the Reviewer on the prolonged recruitment, we have clarified in the study progression paragraph in the Results section that, “The recruitment continued for six months as there was uncertainty as to whether the decline in eligible patients would become permanent, however, the low inclusion rate due to the organizational changes ultimately led to the discontinuation of the study “(page 11, first paragraph). Also note, as we point to in the manuscript, that the trial highlights the difficulty in recruiting participants in this early stage after exposure to trauma. Problems in recruiting have occurred also in other similar trials testing early intervention for patients who experienced trauma, with a low proportion of eligible participants among patients [1-3]. For example, the more recent trial by the Rothbaum et al research group investigating early provided prolonged exposure was terminated before reaching the target number of participants, leaving the study underpowered [2]. We suggest that an important lesson is that this is a factor to consider overall when planning early intervention studies and clinical routines. 

7. Check the ‘tense’ of the entire text. 

Response: We have re-read the manuscript with this in mind and adjusted the tense accordingly, and made additional language edits – we hope that the language now is up to standard. 

8. In my considered opinion [despite the fact which is highly appreciable that tools used (or proposed to be used) are very appropriate] whatever authors want to communicate [definitely important to communicate], could be communicated in just a letter-to-editor or brief communication (and not a full-length paper/article).

Response: We have had extensive discussions in our research group about how to communicate the findings from this study. We agree that the lessons learned could be summarized rather briefly. However, our conclusion is that a full-length manuscript is the best option. A brief communication has its advantages but it would also mean that we would have to exclude many important aspects of the trial (e.g. the rigorous study procedures, both feasibility efficacy and process data, in-depth discussion about organizational aspects and ideas for future research and clinical policy for early interventions for trauma). 

Reviewer #2: 

1. This is an important study that describes an early intervention for PTSD. Although the study was not completed, the results are important in this field given the paucity of studies. 

Response: We thank Reviewer #2 for pointing out the importance of these types of studies.

2. The results and methodology are not completely in sync in terms of previous psychological trauma and psychiatric disorder. How were these assessed? Were participants with current PTSD (ie relating to a previous event) excluded from the study? If not, why not?

Response: We agree with Reviewer #2 that this needs more clarification. As previously shown, exposure to psychological trauma is highly frequent and as much as one-third of the general population will during their life-time experience at least four potentially traumatic events [4]. As reported in the manuscript (table 1, page 12) we assessed history of traumatic events and has clarified that we used a structured assessment interview for this purpose (page 5 under Procedures, first paragraph). Previous traumatic events were not an exclusion criterion. However, a limitation of our study was that we did not assess for current psychiatric disorders and we do not know how many participants that might have fulfilled criteria for PTSD at baseline – this has been added to the limitations (page 18, first paragraph). The clinicians in the current study were very careful to assess the current index trauma (i.e. symptoms that occurred after the incident that lead to the emergency department visit) when scoring the CAPS-5. Participants who reported distress due to exposure to another previous traumatic event received an additional visit to assess symptoms related to this particular event (this extra visit occurred only in one case). Participants were also explicitly instructed to have the current index event in mind when completing the self-rated PCL-5 questionnaire. We have clarified that in the manuscript (page 6, under Measures, first paragraph). 

3. Please provide more information regarding the follow up assessments - were those who attended follow up different from those who did not?

Response: Reviewer #2 raises an important question. Given the low number of recruited participants we did not conduct any statistical analysis. At the 2 months-follow up there was no gender difference in attendance, but at 6-months follow up we saw a small difference, 30 % (6/20) of the recruited women came back and 41% (5/12) of the recruited men. We have added this information on page 14, last paragraph. 

The authors mention sleep but this is not elaborated upon in the discussion given the literature regarding memory consolidation and sleep this is important.

Response: We thank the reviewer for this suggestion, and we elaborate briefly on the role of sleep in early interventions for trauma in the discussion (page 16, last paragraph).

4. The authors do not cover the general field of early intervention of PTSD. The results of the few studies that have been carried out in the first 72 hours, taken together, are not particularly encouraging. Most of the studies of early intervention have concluded that it is most effective with symptomatic survivors, not whole populations, and the few studies examining very early (ER based) interventions are not consistent. This is not reflected in the manuscript, neither in the introduction "One way to decrease the prevalence of PTSD is to intervene before the disorder develops" nor in the discussion, where the recommendation for the use of Internet based interventions or memory consolidation is made without examining the studies that do not support these results. The complexity of this issue should be discussed properly. Additionally the inclusion of participants more than 72 hours after the event raises questions about the use of memory consolidation that should be clarified.

Response: We agree with Reviewer #2 that this issue needs to be clearer in the manuscript and we have elaborated on the current literature in the Discussion (page 17, first paragraph). We have also added to the introduction the findings from a recent study on early intervention (page 3, second paragraph). 

5. The authors state that a number of reasons led to the premature end to the study, but only mention one.

Response: We thank Reviewer #2 for making us aware of that. The reason for the termination of the trial was one, namely organizational changes in the recruiting site that affected the number of eligible participants in a negative way. We have clarified that in the manuscript (page 4, first paragraph and in the results section).

1. Iyadurai L, Blackwell SE, Meiser-Stedman R, Watson PC, Bonsall MB, Geddes JR, et al. Preventing intrusive memories after trauma via a brief intervention involving Tetris computer game play in the emergency department: a proof-of-concept randomized controlled trial. Molecular psychiatry. 2018;23(3):674-82. Epub 2017/03/30. doi: 10.1038/mp.2017.23. PubMed PMID: 28348380; PubMed Central PMCID: PMCPMC5822451.

2. Maples-Keller JL, Post LM, Price M, Goodnight JM, Burton MS, Yasinski CW, et al. Investigation of optimal dose of early intervention to prevent posttraumatic stress disorder: A multiarm randomized trial of one and three sessions of modified prolonged exposure. Depression and anxiety. 2020. Epub 2020/04/06. doi: 10.1002/da.23015. PubMed PMID: 32248637.

3. Rothbaum BO, Kearns MC, Price M, Malcoun E, Davis M, Ressler KJ, et al. Early intervention may prevent the development of posttraumatic stress disorder: a randomized pilot civilian study with modified prolonged exposure. Biol Psychiatry. 2012;72(11):957-63. Epub 2012/07/07. doi: 10.1016/j.biopsych.2012.06.002. PubMed PMID: 22766415; PubMed Central PMCID: PMCPMC3467345.

4. Benjet C, Bromet E, Karam EG, Kessler RC, McLaughlin KA, Ruscio AM, et al. The epidemiology of traumatic event exposure worldwide: results from the World Mental Health Survey Consortium. Psychol Med. 2016;46(2):327-43. Epub 2015/10/30. doi: 10.1017/s0033291715001981. PubMed PMID: 26511595; PubMed Central PMCID: PMCPMC4869975.

---

## [Decision Letter · Decision Letter 1]

30 Dec 2020

PONE-D-20-01989R1

Prevention of Post-Traumatic Stress Disorder: Lessons Learned from a Terminated RCT of Prolonged Exposure

PLOS ONE

Dear Dr. Bragesjo,

Thank you for submitting your manuscript to PLOS ONE. After careful consideration, we feel that it has merit but does not fully meet PLOS ONE’s publication criteria as it currently stands. Therefore, we invite you to submit a revised version of the manuscript that addresses the points raised during the review process.

We look forward to receiving your revised manuscript.

Kind regards,

Negar Fani, PhD

Academic Editor

PLOS ONE

Additional Editor Comments (if provided):

Thank you for the opportunity to evaluate this manuscript. We apologize for the delays in the review process. Reviewers have found many merits to this manuscript but have also outlined some necessary changes (see comments, below) that will need to be addressed before this is suitable for publication in PLOS ONE.

Reviewers' comments:

Reviewer's Responses to Questions

7. PLOS authors have the option to publish the peer review history of their article (what does this mean?). If published, this will include your full peer review and any attached files.

6. Review Comments to the Author

Reviewer #4: The current manuscript reports data from a terminated RCT of a brief trauma focused intervention aimed at preventing the onset of posttraumatic stress disorder. The RTC was terminated due to structural changes within the hospital that led to low recruitment rates. The authors provide recommendations for improving recruitment and delivery of early interventions for PTSD in the context of issues that were encountered in the present study. The original design of the study included many strengths such as the inclusion of an active control therapy, adaptation of an existing evidenced based treatment for PTSD, and longer follow up of participants than previous studies. The authors have largely addressed the concerns of the previous reviewers, however, a few issues remain:

1. The current study reports two main methodological issues: low recruitment rates (largely due to hospital infrastructure) and high attrition (not due to hospital reorganizational changes). The authors primarily provide recommendations for addressing the first issue, but not the second. It would be helpful is recommendations for each of these issues were more specifically specified.

2. With regards to recruitment and retention rates, it is unclear how these rates compare to previous studies that have used similar methods. Are the issues specific to studies recruiting individuals with trauma? Are the issues specific to the hospital at which the study took place?

3. More detail could be provided about the control psychotherapy condition, such as the content that was delivered at each treatment session.

4. Regarding comment #4 or original reviewer #2, I agree that more background on early intervention for PTSD as well as internet delivery could be provided. Some of these details are already included in the Supplement research plan, but could be added to the main text. For example, the authors could note issues with previous brief interventions (e.g. Debriefing). A fuller discussion of the timing chosen for the present study and recommendations could similarly be helpful. It is not clear how treatment within 72 hours addresses the proposed theory of memory consolidation when it is stated that this occurs within 6 hours of an event, and the Rothbaum study used a different time frame (12-24 hours). The authors could discuss strategies and limitations for treating patients that may be hospitalized for acute injuries within such a short time frame. Finally, the authors could add additional research about the use of internet based treatment.

5. Provide the baseline CAPS and PCL-5 scores for both groups.

6. Minor – there is an extra indent on lines 192, 312 and 373

---

## [Author Response · Author response to Decision Letter 1]

14 Jan 2021

PONE-D-20-01989

Dear Editor,

First, we want to thank you and Reviewer #4 for encouraging feedback. We have revised the manuscript accordingly. Below follows a detailed description of the changes made point by point:

Reviewer #4

1. The current manuscript reports data from a terminated RCT of a brief trauma focused intervention aimed at preventing the onset of posttraumatic stress disorder. The RTC was terminated due to structural changes within the hospital that led to low recruitment rates. The authors provide recommendations for improving recruitment and delivery of early interventions for PTSD in the context of issues that were encountered in the present study. The original design of the study included many strengths such as the inclusion of an active control therapy, adaptation of an existing evidenced based treatment for PTSD, and longer follow up of participants than previous studies. The authors have largely addressed the concerns of the previous reviewers.

Response: We thank Reviewer #4 for highlighting the strengths of the original study design. 

2. The current study reports two main methodological issues: low recruitment rates (largely due to hospital infrastructure) and high attrition (not due to hospital reorganizational changes). The authors primarily provide recommendations for addressing the first issue, but not the second. It would be helpful is recommendations for each of these issues were more specifically specified. 

Response: As Reviewer #4 points out, our main issues in the study were the low recruitment rate and the high data attrition levels. We have clarified our recommendations to solves these issues in further studies in the Discussion section (eg. page 17, second paragraph). All changes have been highlighted in red. 

3. More detail could be provided about the control psychotherapy condition, such as the content that was delivered at each treatment session.

Response: We have clarified the content of the control psychotherapy condition in the Control group section (page 11, first paragraph).

4. With regards to recruitment and retention rates, it is unclear how these rates compare to previous studies that have used similar methods. Are the issues specific to studies recruiting individuals with trauma? Are the issues specific to the hospital at which the study took place?

Response: Important point raised by Reviewer #4. This is indeed a common problem also seen in other trials on early interventions after trauma exposure [1-3] and we have clarified this in the Discussion (page 18, first paragraph). 

5. Regarding comment #4 or original reviewer #2, I agree that more background on early intervention for PTSD as well as internet delivery could be provided. Some of these details are already included in the Supplement research plan, but could be added to the main text. For example, the authors could note issues with previous brief interventions (e.g. Debriefing). A fuller discussion of the timing chosen for the present study and recommendations could similarly be helpful. It is not clear how treatment within 72 hours addresses the proposed theory of memory consolidation when it is stated that this occurs within 6 hours of an event, and the Rothbaum study used a different time frame (12-24 hours). The authors could discuss strategies and limitations for treating patients that may be hospitalized for acute injuries within such a short time frame. Finally, the authors could add additional research about the use of internet based treatment. 

Response: We agree with Reviewer #4 and we have added a section in the Introduction on debriefing and a short update on the current status on early interventions after trauma (page 3, second paragraph). We have also clarified the reason for the time-criterium of 72 hours used in the Rothbaum et al (2012) trial [1] (page 4, first paragraph). Our aim with our study was to replicate and extend these findings and we therefore used the same time-criterium for inclusion (page 4, second paragraph). In the Discussion, we also discuss strategies and limitations for providing hospitalized trauma victims within such a short time frame (page 18, second paragraph) and the use of internet-based treatment (page 17, second paragraph). 

6. Provide the baseline CAPS and PCL-5 scores for both groups.

Response: Baseline symptom levels for PTSD was not assessed in this current trial due to the long recall time of CAPS-5 and PCL-5 (CAPS-5 is to be used at the earliest one month after exposure to trauma). The only trauma-related baseline measure used in this study was the Immediate Stress Reaction Checklist (see table 1, page 14).

6. Minor – there is an extra indent on lines 192, 312 and 373. 

Response: Adjusted accordingly. 

1. Rothbaum BO, Kearns MC, Price M, Malcoun E, Davis M, Ressler KJ, et al. Early intervention may prevent the development of posttraumatic stress disorder: a randomized pilot civilian study with modified prolonged exposure. Biol Psychiatry. 2012;72(11):957-63. Epub 2012/07/07. doi: 10.1016/j.biopsych.2012.06.002. PubMed PMID: 22766415; PubMed Central PMCID: PMCPMC3467345.

2. Maples-Keller JL, Post LM, Price M, Goodnight JM, Burton MS, Yasinski CW, et al. Investigation of optimal dose of early intervention to prevent posttraumatic stress disorder: A multiarm randomized trial of one and three sessions of modified prolonged exposure. Depression and anxiety. 2020. Epub 2020/04/06. doi: 10.1002/da.23015. PubMed PMID: 32248637.

3. Iyadurai L, Blackwell SE, Meiser-Stedman R, Watson PC, Bonsall MB, Geddes JR, et al. Preventing intrusive memories after trauma via a brief intervention involving Tetris computer game play in the emergency department: a proof-of-concept randomized controlled trial. Molecular psychiatry. 2018;23(3):674-82. Epub 2017/03/30. doi: 10.1038/mp.2017.23. PubMed PMID: 28348380; PubMed Central PMCID: PMCPMC5822451.

---

## [Decision Letter · Decision Letter 2]

19 Feb 2021

PONE-D-20-01989R2

Prevention of Post-Traumatic Stress Disorder: Lessons Learned from a Terminated RCT of Prolonged Exposure

PLOS ONE

Dear Dr. Bragesjö,

Thank you for submitting your manuscript to PLOS ONE. After careful consideration, we feel that it has merit but does not fully meet PLOS ONE’s publication criteria as it currently stands. Therefore, we invite you to submit a revised version of the manuscript that addresses the points raised during the review process.

We look forward to receiving your revised manuscript.

Kind regards,

Negar Fani, PhD

Academic Editor

PLOS ONE

Additional Editor Comments (if provided):

We appreciate the time taken to revise this manuscript. Although there are noted improvements, it would be informative to include between-group statistics for treatment outcomes and related interpretation of any observed differences. Please see my comments on this and other issues to address, noted below:

Abstract: either include all section headings (background, method, results, conclusions) or no headings/paragraph form.

Objective: the word “of” is omitted in “three sessions … PE”

Results: should be “two-month and six-month assessments”

Figures: In general, resolution is poor, such that the labels are unreadable in Figure 3, please change to high resolution images

Figure 2 What is described as the control condition is described as placebo here, please be consistent with naming

Introduction:

“the intervention seemed to mitigate genetic predisposition to PTSD” is misleading, as psychotherapy cannot “mitigate” genetics—please rephrase to “reduce risk for PTSD development”.

Method:

Control group: line 229, should be “instill hope”

Table 1 should be referenced in the manuscript and a column added for relevant statistics and indication of any significant differences between intervention groups (Chi square, Fisher’s z, t statistic)

Table 2: given that 4 participants dropped out, 2 from each intervention, and the starting number is 32, shouldn’t the correct n be 13 and 15, respectively? Please also indicate the correct sample size for each group at 2 and 6 month follow up in the table, which is significantly smaller than the 32 that is listed in the top row.

Results: Even though PTSD was not assessed at baseline and sample size is small, please provide statistics both in Table 2 (in a column at the right) and in the text for all outcome measures. Although the numbers for each group are small, it is possible to examine differences in treatment outcomes, particularly with assessments that were also given at baseline (e.g., MADRS-S – repeated measures ANOVA

Reviewers' comments:

Reviewer's Responses to Questions

**Comments to the Author**

1. If the authors have adequately addressed your comments raised in a previous round of review and you feel that this manuscript is now acceptable for publication, you may indicate that here to bypass the “Comments to the Author” section, enter your conflict of interest statement in the “Confidential to Editor” section, and submit your "Accept" recommendation.

Reviewer #4: All comments have been addressed

2. Is the manuscript technically sound, and do the data support the conclusions?

Reviewer #4: Yes

3. Has the statistical analysis been performed appropriately and rigorously? 

Reviewer #4: N/A

4. Have the authors made all data underlying the findings in their manuscript fully available?

Reviewer #4: Yes

5. Is the manuscript presented in an intelligible fashion and written in standard English?

Reviewer #4: Yes

6. Review Comments to the Author

Reviewer #4: The present manuscript is a revised submission that provides recommendations for providing brief interventions for trauma patients to prevent the onset of PTSD. The authors have addressed the reviewers' comments.

7. PLOS authors have the option to publish the peer review history of their article (what does this mean?). If published, this will include your full peer review and any attached files.

Reviewer #4: No

---

## [Author Response · Author response to Decision Letter 2]

29 Mar 2021

PONE-D-20-01989R2

Dear Editor,

We are grateful for the valuable feedback from you and Reviewer #4 and we now submit our revised manuscript for review. Below you will find a detailed description of the changes made point by point:

Reviewer #4

1. The present manuscript is a revised submission that provides recommendations for providing brief interventions for trauma patients to prevent the onset of PTSD. The authors have addressed the reviewers' comments.

Response: We thank Reviewer #4 for their helpful comments and positive judgement of our revised manuscript. 

Editor Comments

1. We appreciate the time taken to revise this manuscript. Although there are noted improvements, it would be informative to include between-group statistics for treatment outcomes and related interpretation of any observed differences. Please see my comments on this and other issues to address, noted below:

Response: We thank the Editor encouraging feedback. While we appreciate the editor´s feedback, we respectfully disagree on adding statistical tests. The study is not powered to detect any statistically significant changes between groups with the small sample size we ended up with. We fear that adding these analyses post hoc would be potentially misleading. Adding p-values might deflect attention from the actual size of an effect. We note that the reviewers are content with the way we have described the collected data which we feel supports our concern. 

2. Abstract: either include all section headings (background, method, results, conclusions) or no headings/paragraph form.

Response: We have excluded the section headings in the abstract (page 2). 

3. Objective: the word “of” is omitted in “three sessions … PE” Results: should be “two-month and six-month assessments”

Response: We have proof read the manuscript and added the word “of” on page 2 in the Abstract section and have used the word two-month and six-month assessment consequently. All changes are highlighted in red through the manuscript. 

4. Figures: In general, resolution is poor, such that the labels are unreadable in Figure 3, please change to high resolution images. Figure 2 What is described as the control condition is described as placebo here, please be consistent with naming.

Response: We have updated the figures to a higher resolution. We have also changed the wording in figure 2 to “control group”. 

5. Introduction: “the intervention seemed to mitigate genetic predisposition to PTSD” is misleading, as psychotherapy cannot “mitigate” genetics—please rephrase to “reduce risk for PTSD development”.

Response: We have rephrased the sentence to “reduce risk for PTSD development” (page 4, first paragraph). 

6. Method: Control group: line 229, should be “instill hope”

Response: We have corrected to phrase to be “instill hope” (page 11, first paragraph). 

7. Table 1 should be referenced in the manuscript and a column added for relevant statistics and indication of any significant differences between intervention groups (Chi square, Fisher’s z, t statistic)

Response: Table 1 is now referenced on page 13, second paragraph. As for the statistical tests, we refer to our response to comment no. 1. 

8. Table 2: given that 4 participants dropped out, 2 from each intervention, and the starting number is 32, shouldn’t the correct n be 13 and 15, respectively? Please also indicate the correct sample size for each group at 2 and 6 month follow up in the table, which is significantly smaller than the 32 that is listed in the top row.

Response: We thank the editor for pointing out that the number of respondents is unclear in table 2. We have redesigned table 2 in order to clarify the number of respondents at each timepoint. 

9. Results: Even though PTSD was not assessed at baseline and sample size is small, please provide statistics both in Table 2 (in a column at the right) and in the text for all outcome measures. Although the numbers for each group are small, it is possible to examine differences in treatment outcomes, particularly with assessments that were also given at baseline (e.g., MADRS-S – repeated measures ANOVA

Response: Please see answer to point 1. We agree that such analysis could be interesting, but fear that they would be misleading. We believe that it is possible to examine the differences by looking at the data points provided for the outcome measures and exercising judgment on the absolute differences across time points.

---

## [Editor Report · Decision Letter 3]

6 May 2021

Prevention of Post-Traumatic Stress Disorder: Lessons Learned from a Terminated RCT of Prolonged Exposure

PONE-D-20-01989R3

Dear Dr. Bragesjo,

We’re pleased to inform you that your manuscript has been judged scientifically suitable for publication and will be formally accepted for publication once it meets all outstanding technical requirements.

Kind regards,

Negar Fani, PhD

Academic Editor

PLOS ONE

Additional Editor Comments (optional):

Please upload higher resolution Figure 3
---

## [Editor Report · Acceptance letter]

15 May 2021

PONE-D-20-01989R3 

Prevention of Post-Traumatic Stress Disorder: Lessons Learned from a Terminated RCT of Prolonged Exposure 

Dear Dr. Bragesjö:

I'm pleased to inform you that your manuscript has been deemed suitable for publication in PLOS ONE. Congratulations! Your manuscript is now with our production department. 

Kind regards, 

on behalf of

Dr. Negar Fani 

Academic Editor

PLOS ONE